# Development and validation of predictive models for diabetic retinopathy using machine learning

Penglu Yang[1], Bin Yang[2]*

1 The First Clinical School & Union Hospital, Tongji Medical College, Huazhong University of Science and Technology, Wuhan, Hubei, China, 2 Health Management Center, Union Hospital, Tongji Medical College, Huazhong University of Science and Technology, Wuhan, Hubei, China

* yangbin888@hust.edu.cn

## Abstract

### Objective

This study aimed to develop and compare machine learning models for predicting diabetic retinopathy (DR) using clinical and biochemical data, specifically logistic regression, random forest, XGBoost, and neural networks.

### Methods

A dataset of 3,000 diabetic patients, including 1,500 with DR, was obtained from the National Population Health Science Data Center. Significant predictors were identified, and four predictive models were developed. Model performance was assessed using accuracy, precision, recall, F1-score, and area under the curve (AUC).

### Results

Random forest and XGBoost demonstrated superior performance, achieving accuracies of 95.67% and 94.67%, respectively, with AUC values of 0.991 and 0.989. Logistic regression yielded an accuracy of 76.50% (AUC: 0.828), while neural networks achieved 82.67% accuracy (AUC: 0.927). Key predictors included 24-hour urinary microalbumin, HbA1c, and serum creatinine.

### Conclusion

The study highlights random forest and XGBoost as effective tools for early DR detection, emphasizing the importance of renal and glycemic markers in risk assessment. These findings support the integration of machine learning models into clinical decision-making for improved patient outcomes in diabetes management.

## Introduction

Diabetic retinopathy (DR) is one of the most prevalent and serious complications of diabetes, affecting approximately one-third of diabetic patients worldwide and serving as a leading

**Data availability statement:** The research data utilized in this study is sourced from a publicly available dataset, accessible at the following URL: https://www.ncmi.cn//phda/dataDetails. do?id=CSTR:A0006.11.A0005.201905.000282. The minimal dataset used in our analysis has also been provided in the Supporting information files accompanying the manuscript.

**Funding:** The author(s) received no specific funding for this work.

**Competing interests:** The authors have declared that no competing interests exist.

cause of blindness among working-age adults. Early detection of DR is crucial, as timely intervention can prevent progression to more advanced stages, such as proliferative DR or diabetic macular edema, both of which are associated with significant vision loss. However, DR is frequently asymptomatic during its early stages, underscoring the importance of regular screening for the identification of at-risk patients [1].

Traditional screening methods for DR depend on retinal imaging and manual examinations conducted by trained specialists, which can be resource-intensive and may not be feasible in areas with limited healthcare access. Consequently, there is an increasing interest in employing machine learning (ML) models to aid in the early detection of DR by analyzing routine clinical and biochemical data [2,3]. These models have the potential to alleviate the burden on healthcare systems by automating the risk stratification process, thereby facilitating the earlier and more efficient identification of patients who require further ophthalmologic evaluation.

Despite the expanding body of research on machine learning applications in DR prediction, few studies have thoroughly compared multiple models using a large dataset to ascertain which model provides the optimal balance of predictive performance and clinical applicability. This study aims to address this gap by evaluating the performance of four machine learning models—logistic regression, random forest, XGBoost, and neural networks—in predicting DR, utilizing clinical and biochemical data from a substantial national dataset. By assessing each model's accuracy, precision, recall, and area under the curve (AUC), this study seeks to identify the most effective machine learning approach for DR prediction and to explore its potential for integration into clinical practice.

## Patients and methods

### Study population

This study is a retrospective cross-sectional analysis that utilizes clinical data from the National Population Health Data Center's PHDA Diabetes Complications Data Set on May 6th, 2024 (https://www.ncmi.cn//phda/dataDetails.do?id=CSTR:A0006.11. A0005.201905.000282). This dataset includes clinical information on 3,000 diabetic patients, of which 1,500 were diagnosed with DR. The remaining 1,500 patients had diabetes without DR, serving as the control group.

### Ethics declarations

The data used in this research comes from a public database, specifically the National Population Health Data Center's PHDA Diabetes Complications Data Set. The patients' information in this database has been anonymized, eliminating the need for ethical approval and informed consent from the patients involved.

### Data collection

The data extracted from the dataset included both demographic information and biochemical parameters. Demographic data such as age, gender, and clinical characteristics including blood pressure, fasting blood glucose, HbA1c, serum creatinine, and 24-hour urinary microalbumin were collected. Prior to model development, the dataset underwent cleaning to handle missing or inconsistent values, and preprocessing to normalize features where appropriate.

### Model development and evaluation

To predict the presence of DR, four machine learning models were meticulously crafted and evaluated: logistic regression, random forest, XGBoost, and neural networks. The dataset was strategically divided into training (80%) and testing (20%) sets to ensure robust validation.

Logistic Regression: This model incorporated L2 regularization for bias reduction, employing the lbfgs optimizer with a cap on iterations set at 100 to ensure convergence.

Random Forest: Comprising 100 decision trees, this model leveraged Gini impurity as the criterion for node splitting, enhancing its ability to capture complex patterns.

XGBoost: Utilizing a tree-based boosting algorithm, this model was configured with 100 estimators and a learning rate of 0.1, optimizing its predictive power.

Neural Networks: A single hidden layer with 100 neurons was employed, activated by the ReLU function, and optimized via the adam optimizer to handle non-linear relationships effectively.

The performance of each model was rigorously assessed using accuracy, precision, recall, F1-score, and the area under the curve (AUC) of the ROC curve. These metrics were calculated for both the training and testing datasets, providing a comprehensive evaluation of each model's predictive capabilities for diabetic retinopathy. Fine-tuning of model parameters was achieved through cross-validation and grid search, ensuring optimal performance.

## Statistical methods

Before performing quantitative data analysis, normality was assessed using the Jarque-Bera test, and homogeneity of variance was checked. For variables following a normal distribution, data were presented as mean ± standard deviation (SD), while non-normally distributed variables were expressed as median (interquartile range, IQR). For normally distributed continuous variables, independent samples t-tests were used to compare means between the DR and non-DR groups. For variables not normally distributed or with unequal variances, Mann-Whitney $U$ tests were employed. Categorical variables were summarized as $n$ (%) and compared between groups using the chi-squared ($\chi^2$) test. A two-tailed $p$ value < 0.05 was considered statistically significant for all tests.

## Results

### Comparison of baseline data

As shown in Table 1. Patients in the DR group were significantly younger than those in the non-DR group, with mean ages of 56.59 ± 10.94 years and 58.99 ± 11.24 years, respectively ($p < 0.001$). This suggests that DR may progress more aggressively in younger individuals. Additionally, systolic and diastolic blood pressures were both notably higher in the DR group (142.36 ± 21.42 mmHg and 81.97 ± 11.81 mmHg) compared to the non-DR group (135.00 ± 19.87 mmHg and 78.96 ± 11.86 mmHg), with $p < 0.001$ for both measures.

Glycemic markers also showed significant differences. Patients in the DR group exhibited higher fasting blood glucose (8.70 ± 4.08 mmol/L) and HbA1c levels (8.15 ± 1.87%) than those in the non-DR group, with mean fasting glucose of 8.20 ± 3.69 mmol/L and HbA1c of 7.44 ± 1.52% ($p = 0.001$ and $p < 0.001$, respectively), suggesting that poorer glycemic control is associated with DR. Renal function indicators were also significantly different; serum creatinine and 24-hour urinary microalbumin levels were elevated in the DR group (78.75 [60.31–122.60] μmol/L and 1.46 ± 1.20 mg/L) compared to the non-DR group (69.55 [58.51–82.41] μmol/L and 0.95 ± 0.56 mg/L), with $p < 0.001$.

Lipid metabolism markers also varied significantly between groups. Serum uric acid levels were higher in the DR group (329.10 [269.82–398.90] μmol/L) than in the non-DR group (310.75 [251.3–372.6] μmol/L), and total cholesterol levels were similarly elevated in the DR group (4.53 [3.28–5.43] mmol/L) versus the non-DR group (4.43 [3.27–5.12] mmol/L), both with $p < 0.001$.

**Table 1. Baseline characteristics of the study population.**

| Variable | Non-DR Group (*n* = 1500) | DR Group (*n* = 1500) | *t*/Mann-Whitney *U* | *P* value |
|---|---|---|---|---|
| Age (years) | 58.99 ± 11.24 | 56.59 ± 10.94 | 5.931 | < 0.001 |
| Body Mass Index (kg/m²) | 26.19 ± 3.62 | 26.40 ± 3.26 | −1.652 | 0.099 |
| Systolic Blood Pressure (mmHg) | 135.00 ± 19.87 | 142.36 ± 21.42 | −9.754 | < 0.001 |
| Diastolic Blood Pressure (mmHg) | 78.96 ± 11.86 | 81.97 ± 11.81 | −6.981 | < 0.001 |
| Fasting Blood Glucose (mmol/L) | 8.20 ± 3.69 | 8.70 ± 4.08 | −3.453 | 0.001 |
| HbA1c (%) | 7.44 ± 1.52 | 8.15 ± 1.87 | −11.385 | < 0.001 |
| 24h Urinary Microalbumin (mg/L) | 0.95 ± 0.56 | 1.46 ± 1.20 | −14.764 | < 0.001 |
| Serum Creatinine (μmol/L) | 69.55 (58.51–82.41) | 78.75 (60.31–122.60) | −10.251 | < 0.001 |
| Serum Uric Acid (μmol/L) | 310.75 (251.3–372.6) | 329.10 (269.82–398.90) | −5.634 | < 0.001 |
| Total Cholesterol (mmol/L) | 4.43 (3.27–5.12) | 4.53 (3.28–5.43) | −4.602 | < 0.001 |
| Glycated Serum Protein (%) | 223.87 ± 55.03 | 228.57 ± 73.03 | −1.992 | 0.046 |
| Urine Protein Creatinine Ratio (mg/mmol) | 56.65 (35.32,56.62) | 194.11 (18.01,206.02) | −18.067 | < 0.001 |
| Blood Urea (mmol/L) | 5.39 (4.44,6.62) | 6.39 (5.02,9.02) | −12.719 | < 0.001 |
| Low-Density Lipoprotein Cholesterol (mmol/L) | 2.72 (2.04,3.24) | 2.82 (2.25,3.56) | −5.226 | < 0.001 |
| Total Bilirubin (μmol/L) | 10.00 (7.51,13.72) | 8.65 (6.04,12.22) | −8.769 | < 0.001 |
| Direct Bilirubin (μmol/L) | 3.20 (2.34,4.44) | 2.30 (1.52,3.52) | −13.941 | < 0.001 |
| Indirect Bilirubin (μmol/L) | 6.80 (4.91,9.54) | 6.20 (4.31,8.71) | −5.164 | < 0.001 |
| Alanine Aminotransferase (U/L) | 20.70 (13.84,30.81) | 16.30 (11.81,23.71) | −10.093 | < 0.001 |
| Aspartate Aminotransferase (U/L) | 17.20 (13.59,22.95) | 15.50 (12.61,20.02) | −8.432 | < 0.001 |
| Gamma-Glutamyl Transferase (U/L) | 28.90 (18.44,50.45) | 22.90 (15.91,34.94) | −10.115 | < 0.001 |
| Prothrombin Activity (%) | 100.00 (91.40,108.04) | 100.25 (92.05,110.40) | −2.272 | 0.023 |
| Prothrombin Time (t) | 13.41 ± 4.11 | 13.10 ± 3.47 | 2.175 | 0.030 |
| Alkaline Phosphatase (U/L) | 70.10 (57.6,85.4) | 67.80 (55.81,82.81) | −3.157 | 0.002 |
| Lactate Dehydrogenase (U/L) | 154.95 (135.81,181.72) | 165.00 (142.51,194.81) | −6.547 | < 0.001 |
| Fibrinogen (g/L) | 3.31 (2.85,4.14) | 3.68 (3.04,4.84) | −7.722 | < 0.001 |
| Serum Lipase (U/L) | 165.11 (106.52,165.11) | 166.49 (118.11,166.52) | −14.507 | < 0.001 |
| Hemoglobin (g/L) | 136.30 ± 21.40 | 127.11 ± 23.78 | 11.131 | < 0.001 |
| Hematocrit (L/L) | 0.40 ± 0.06 | 0.37 ± 0.07 | 12.714 | < 0.001 |
| Total Protein (g/L) | 67.31 ± 6.56 | 63.66 ± 7.82 | 13.873 | < 0.001 |
| Serum Albumin (g/L) | 40.85 ± 5.14 | 37.92 ± 6.18 | 14.117 | < 0.001 |
| C-Reactive Protein (mg/L) | 1.53 ± 2.67 | 0.85 ± 1.81 | 8.174 | < 0.001 |
| Globulin (g/L) | 26.45 ± 4.94 | 25.74 ± 4.79 | 3.996 | < 0.001 |
| Triglycerides (mmol/L) | 1.63 (1.12,2.32) | 1.55 (1.14,2.34) | −0.528 | 0.598 |
| High-Density Lipoprotein Cholesterol (mmol/L) | 1.06 ± 0.30 | 1.08 ± 0.32 | −1.787 | 0.074 |
| Platelets (10^9/L) | 210.00 (174.40,251.48) | 211.00 (173.10,254.30) | −0.387 | 0.698 |
| Activated Partial Thromboplastin Time (APTT, s) | 35.90 (33.41,38.91) | 36.00 (33.51,38.72) | −0.315 | 0.753 |
| Carbohydrate Antigen199 (U/ml) | 16.56 (9.20,40.31) | 19.18 (11.522.72) | −0.523 | 0.601 |

## Screening of characteristic factors

To identify significant predictors for DR, a multivariate logistic regression analysis was conducted on demographic, biochemical, and clinical factors. This analysis revealed 11 key factors associated with DR risk, each of which demonstrated statistical significance, highlighting their independent contribution to DR prediction while controlling for potential confounders. As shown in Table 2.

Among these predictors, HbA1c and glycated serum protein emerged as two of the strongest glycemic markers, with odds ratios (OR) of 1.476 ($p < 0.001$) and 1.004 ($p < 0.001$),

**Table 2. Multivariate logistic regression analysis of characteristic factors for diabetic retinopathy.**

| Variable | Coefficient (β) | Standard Error | z value | Wald χ² | P value | Odds Ratio (OR) | 95% CI for OR |
|---|---|---|---|---|---|---|---|
| Age (years) | −0.016 | 0.004 | −3.650 | 13.321 | < 0.001 | 0.985 | 0.976–0.993 |
| Diastolic Blood Pressure (mmHg) | 0.014 | 0.004 | 3.431 | 11.770 | 0.001 | 1.014 | 1.006–1.022 |
| Fasting Blood Glucose (mmol/L) | −0.052 | 0.014 | −3.856 | 14.871 | < 0.001 | 0.949 | 0.924–0.975 |
| HbA1c (%) | 0.389 | 0.035 | 11.033 | 121.738 | < 0.001 | 1.476 | 1.377–1.582 |
| Glycated Serum Protein (%) | 0.004 | 0.001 | 4.042 | 16.337 | < 0.001 | 1.004 | 1.002–1.005 |
| 24-hour Urinary Microalbumin (mg/L) | 0.298 | 0.061 | 4.854 | 23.564 | < 0.001 | 1.347 | 1.194–1.519 |
| Hemoglobin (g/L) | 0.040 | 0.01 | 4.078 | 16.629 | < 0.001 | 1.041 | 1.021–1.062 |
| Hematocrit (L/L) | −20.196 | 3.637 | −5.552 | 30.826 | < 0.001 | 0.000 | 0.000–0.000 |
| Aspartate Aminotransferase (U/L) | −0.021 | 0.004 | −4.944 | 24.442 | < 0.001 | 0.979 | 0.971–0.987 |
| Urine Protein Creatinine Ratio (mg/mmol) | 0.009 | 0.001 | 14.521 | 210.851 | < 0.001 | 1.009 | 1.008–1.010 |
| C-Reactive Protein (mg/L) | −0.163 | 0.026 | −6.202 | 38.468 | < 0.001 | 0.849 | 0.807–0.894 |

respectively, emphasizing the importance of long-term glycemic control in mitigating DR risk. Renal markers, including 24-hour urinary microalbumin (OR = 1.347, $p < 0.001$) and urine protein creatinine ratio (OR = 1.009, $p < 0.001$), also showed a strong association with DR, reinforcing the link between kidney function and DR progression.

Other variables, such as diastolic blood pressure (OR = 1.014, $p = 0.001$) and C-reactive protein (CRP) (OR = 0.849, $p < 0.001$), added further insights. Higher diastolic blood pressure was positively associated with DR, suggesting a potential role of hypertension, while elevated CRP, an inflammatory marker, showed an inverse relationship with DR, indicating that inflammation may have complex effects in DR's progression.

Additional factors, such as age (OR = 0.985, $p < 0.001$), hemoglobin (OR = 1.041, $p < 0.001$), and aspartate aminotransferase (AST) (OR = 0.979, $p < 0.001$), contributed meaningfully to the risk profile, offering further context to DR risk. Younger age and elevated hemoglobin levels were associated with higher DR risk, while AST showed a slight protective effect. Collectively, these 11 characteristic factors across glycemic, renal, cardiovascular, and inflammatory domains provide a multifaceted view of DR risk, supporting their inclusion in subsequent predictive model development and validation.

## Development and validation of predictive models

We developed and validated four machine learning models—logistic regression, random forest, XGBoost, and neural networks—to predict the risk of DR based on the 11 characteristic factors identified through multivariate analysis. The dataset was divided into training (80%) and testing (20%) sets, with cross-validation employed to optimize model parameters. Each model's performance was assessed using accuracy, precision, recall, F1-score, and area under the curve (AUC).

As shown in Table 3, tree-based models (Random Forest and XGBoost) exhibited high accuracy and AUC values (close to or above 0.98), indicating their effectiveness in distinguishing DR from non-DR cases. Their balanced precision and recall further underscore their reliability in detecting DR, effectively minimizing both false positives and false negatives, which makes them particularly suitable for early screening programs.

Logistic regression, while exhibiting lower accuracy and AUC compared to tree-based models, retains significant value due to its simplicity and interpretability. This makes it especially advantageous in healthcare settings that prioritize Model transparency to support clinical decision-making.

**Table 3. Performance of predictive models on test set.**

| Model | Accuracy (%) | Precision (%) | Recall (%) | F1-Score | AUC (95% CI) | P value |
|-------|-------------|---------------|-----------|----------|--------------|---------|
| Logistic Regression | 76.50 | 76.79 | 76.50 | 0.763 | 0.828 (0.814–0.843) | < 0.001 |
| Random Forest | 95.67 | 95.89 | 95.67 | 0.957 | 0.991 (0.988–0.995) | < 0.001 |
| XGBoost | 94.67 | 94.85 | 94.67 | 0.947 | 0.989 (0.985–0.994) | < 0.001 |
| Neural Networks | 82.67 | 83.14 | 82.67 | 0.827 | 0.927 (0.918–0.936) | < 0.001 |

The neural network model achieved moderate accuracy and AUC but showed potential in capturing non-linear relationships within the data. This capacity may prove beneficial for more complex datasets, even though its performance was less consistent in terms of precision and recall.

Fig 1 presents the receiver operating characteristic (ROC) curves for all models. Random Forest and XGBoost achieved the highest AUC values (0.991 and 0.989, respectively), followed by the neural network model (AUC = 0.927) and logistic regression (AUC = 0.828). Fig 2 compares the performance metrics of the four models, highlighting the superior accuracy, AUC, precision, and recall of Random Forest and XGBoost. Logistic regression maintained consistent but lower scores across all metrics, while the neural network exhibited competitive accuracy and AUC but underperformed in precision and recall.

## Feature importance analysis

The feature importance analysis was conducted using the random forest and XGBoost models to identify the most influential predictors of DR, which was shown in Table 4, Figs 3 and 4.

In the random forest model, 24-hour urinary microalbumin emerged as the most important predictor, accounting for 35.5% of the importance score. This is consistent with existing literature that highlights the association between renal function and diabetic complications. The XGBoost model placed even greater emphasis on this variable, with an importance score of 63.6%. Such high importance underscores the critical role of kidney health in DR risk assessment.

The urine protein creatinine ratio also showed substantial predictive power, particularly in the random forest model, where it contributed 29.7%. In XGBoost, its importance was lower at 9.4%, but it still remains a significant marker. Glycated serum protein and C-reactive protein followed, with importance scores of 12.4% and 9.4% in random forest, and 9.2% and 3.2% in XGBoost, respectively. These findings indicate that both long-term glycemic control and inflammatory responses are critical factors in determining DR risk.

Other factors, such as glycated hemoglobin, hematocrit, and hemoglobin levels, contributed modestly to the models. Their lower importance scores (2.9% to 2.2%) suggest that while they are relevant, they may not be as impactful as renal and glycemic indicators. Fasting blood glucose, age, aspartate aminotransferase, and diastolic blood pressure all demonstrated minor contributions, with scores ranging from 1.4% to 1.6%.

## Discussion

DR is a leading cause of vision impairment among adults, making early detection critical for effective intervention. This study aimed to enhance DR prediction using machine learning models, specifically comparing the performance of logistic regression, random forest, XGBoost, and neural networks [4–6]. Our findings demonstrated that random forest and XGBoost significantly outperformed logistic regression and neural networks in terms of predictive accuracy, AUC, precision, and recall, establishing these tree-based models as highly

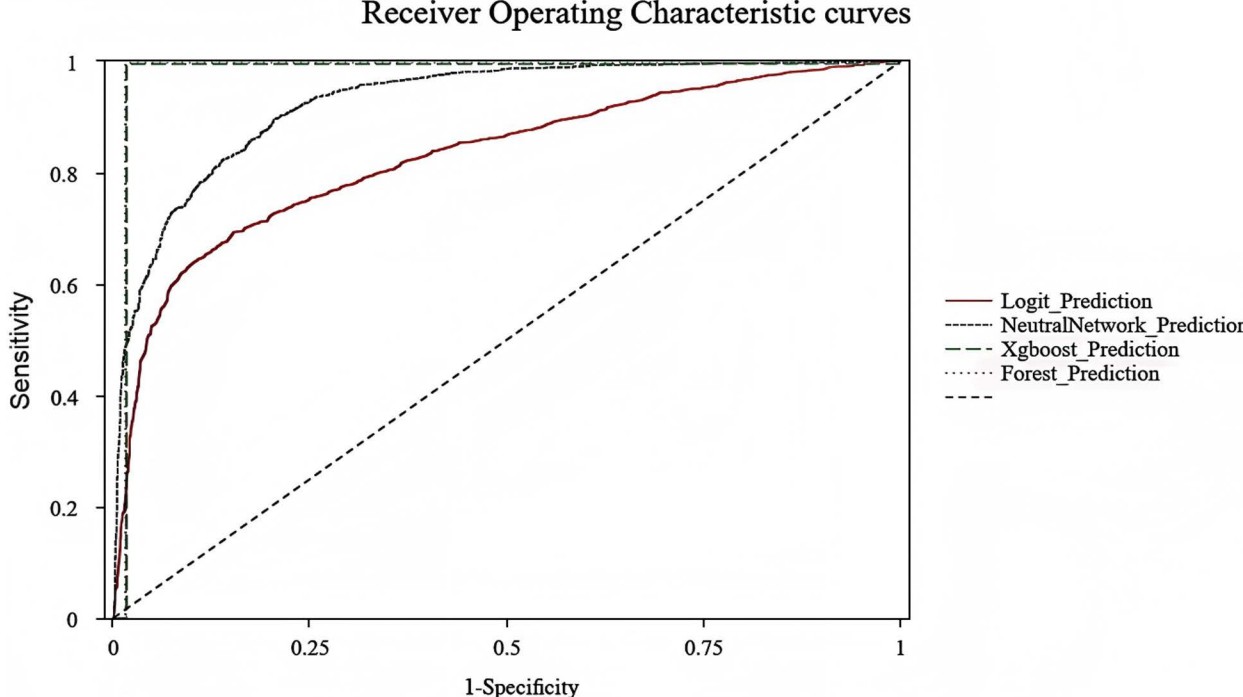

**Fig 1. Receiver operating characteristic (ROC) curves for the four predictive models.** The curves illustrate the sensitivity (true positive rate) versus 1-specificity (false positive rate) for each model. The diagonal dashed line represents the performance of a random classifier. XGBoost and Random Forest models show superior performance with the largest areas under the curve (AUC), indicating better discriminatory power compared to Logistic Regression and Neural Networks.

effective tools for DR prediction. Furthermore, the study identified key predictors, notably 24-hour urinary microalbumin [7], HbA1c, and serum creatinine, which are strongly associated with DR risk [8,9]. These results have potential implications for improving DR screening and underscore the value of machine learning models in clinical decision-making [10,11].

Our results are consistent with previous studies that have demonstrated the effectiveness of tree-based models, particularly random forest and XGBoost, in predicting diabetic complications [12]. Research by Alam et al. reported that random forest and XGBoost achieved high AUC values and accuracy in DR prediction, emphasizing the utility of these models in analyzing complex healthcare data [13]. Additionally, recent work highlights the potential of XGBoost in accurately predicting DR risk based on biochemical and imaging data, further validating our findings [14–16]. The high predictive power and robustness of random forest in handling large datasets are well-documented, and our findings align with these observations, suggesting that random forest may be an optimal choice for DR screening [17]. XGBoost also performed well, with accuracy and AUC values comparable to those of random forest, demonstrating strong classification capabilities and effective handling of variable importance [15]. These findings highlight the utility of tree-based models in healthcare applications where early detection is critical [18,19].

In contrast, the logistic regression model, while achieving moderate accuracy, demonstrated lower performance compared to the tree-based models. Nevertheless, the transparency and interpretability of logistic regression make it an appealing option in clinical settings where simplicity is essential. Clinicians often require understandable models that clearly delineate how variables contribute to predictions, and logistic regression provides a straightforward

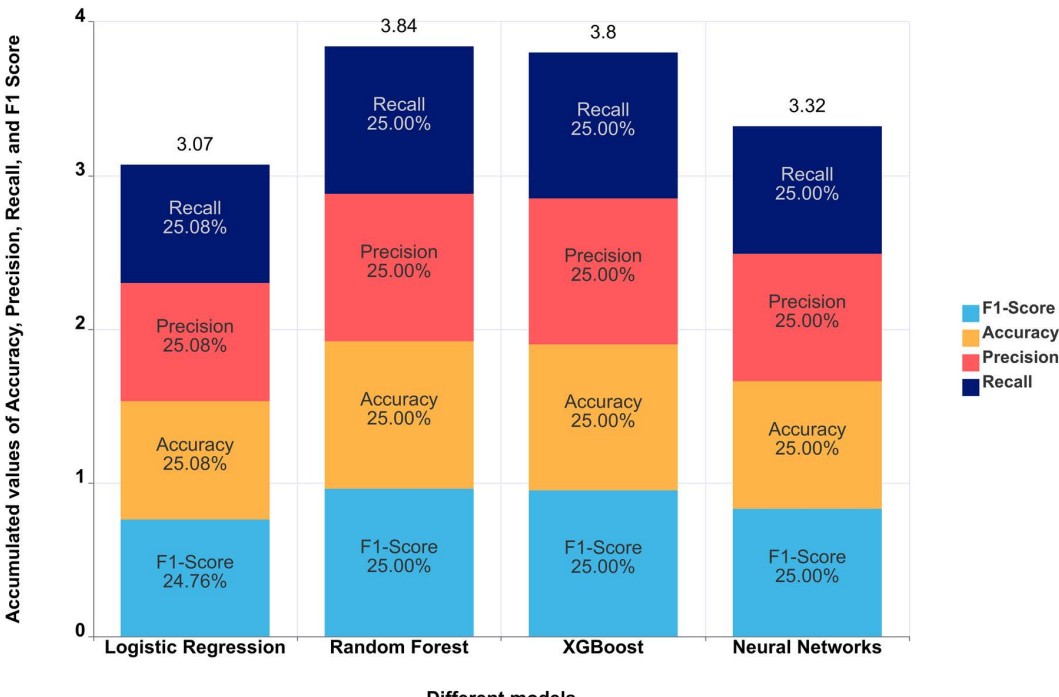

**Fig 2. Bar chart comparing accuracy, precision, recall, and F1-score across four machine learning models.** Each stacked bar represents the accumulated values of accuracy, precision, recall, and F1-score. The Random Forest and XGBoost models demonstrated the highest combined metrics, indicating their superior predictive capabilities. Distinct colors identify each metric: yellow for accuracy, red for precision, dark blue for recall, and light blue for F1-score.

**Table 4. Feature importance for Random Forest and XGBoost Models.**

| Feature | Random Forest Importance | XGBoost Importance | P value |
|---|---|---|---|
| 24h Urinary Microalbumin (mg/L) | 35.5% | 63.6% | < 0.001 |
| Urine Protein Creatinine Ratio (mg/mmol) | 29.7% | 9.4% | < 0.001 |
| Glycated Serum Protein (%) | 12.4% | 9.2% | < 0.001 |
| C-Reactive Protein (mg/L) | 9.4% | 3.2% | 0.002 |
| Glycated Hemoglobin (%) | 2.9% | 4.1% | 0.005 |
| Hematocrit (L/L) | 2.2% | 2.0% | 0.006 |
| Hemoglobin (g/L) | 1.9% | 2.6% | 0.008 |
| Fasting Blood Glucose (mmol/L) | 1.6% | 1.8% | 0.010 |
| Age (years) | 1.5% | 1.4% | 0.011 |
| Aspartate Aminotransferase (U/L) | 1.5% | 1.6% | 0.013 |
| Diastolic Blood Pressure (mmHg) | 1.4% | 1.0% | 0.015 |

interpretation of risk factors, which can facilitate shared decision-making and enhance patient communication [20]. Previous studies, such as that by Yau et al. (2018), have indicated that logistic regression remains valuable for diabetic retinopathy (DR) prediction, particularly in resource-limited environments or when interpretability is prioritized over predictive power.

Although neural networks surpassed logistic regression in accuracy and AUC, they did not achieve the predictive performance of random forest and XGBoost. Neural networks are recognized for their capacity to model complex, non-linear relationships among features, which

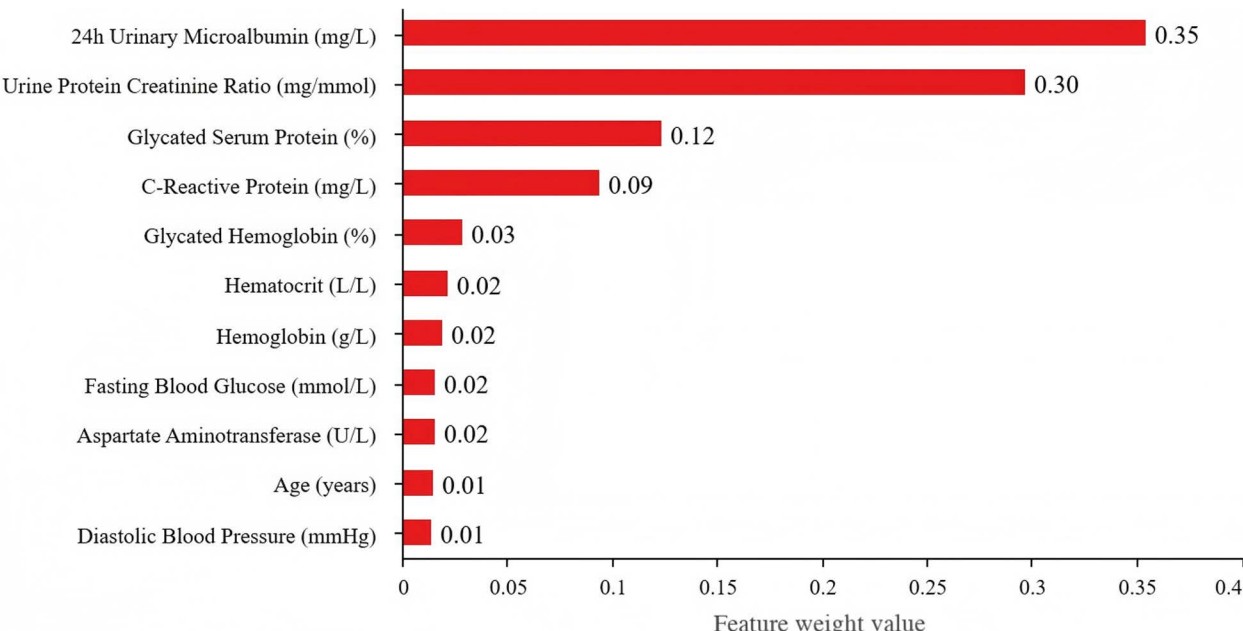

**Fig 3. Feature weight values in the Random Forest model.** This bar chart illustrates the relative importance of features as determined by the Random Forest model. The x-axis represents the weight values, ranging from 0 to 0.4, and the y-axis lists the featuress. '24h Urinary Microalbumin (mg/L)' and 'Urine Protein Creatinine Ratio (mg/mmol)' demonstrate the highest weight values, highlighting their significant contributions to the model's predictions. Other features show progressively lower weights, emphasizing their comparatively lesser importance in the analysis.

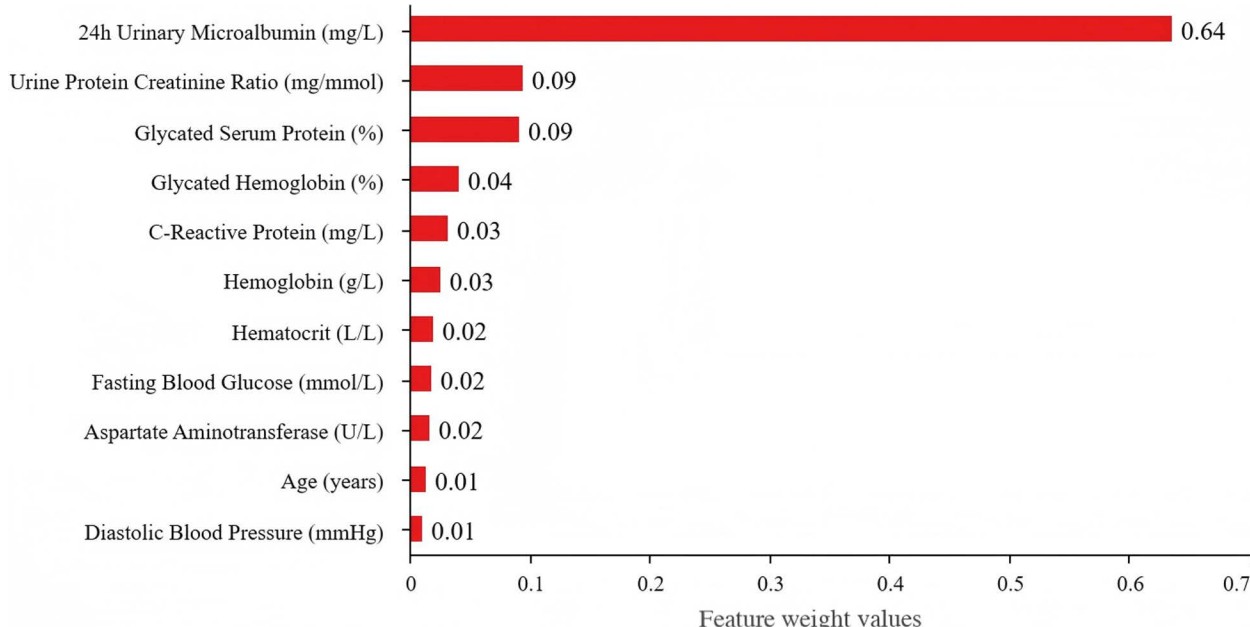

**Fig 4. Feature weight values in the XGBoost Model.** This bar chart illustrates the relative importance of features as determined by the XGBoost Model. The x-axis represents the weight values, ranging from 0 to 0.4, and the y-axis lists the featuress. '24h Urinary Microalbumin (mg/L)' and 'Urine Protein Creatinine Ratio (mg/mmol)' also demonstrate the highest weight values, highlighting their significant contributions to the model's predictions. Other features show progressively lower weights, emphasizing their comparatively lesser importance in the analysis.

can be advantageous in situations where subtle interactions influence DR risk. However, recent publications by Lee et al. suggest that advanced architectures and tuning are necessary to unlock the full potential of neural networks in DR prediction [21]. These findings suggest that while neural networks may capture additional layers of data complexity, their application in DR prediction may necessitate further refinement and optimization to reach the performance levels of tree-based models in this context.

Random forest and XGBoost, noted for their high sensitivity and specificity, show promise as reliable tools for diabetic retinopathy (DR) screening that could be integrated into clinical decision support systems. Notably, the high sensitivity of these models minimizes false negatives, which is crucial in preventing undetected cases of DR that may progress to more severe stages. By incorporating these models into DR screening programs, clinicians could facilitate the early identification of high-risk patients, allowing for timely referrals for ophthalmologic evaluation and early intervention. Furthermore, the feature importance analysis highlights the relevance of renal and glycemic markers—specifically, 24-hour urinary microalbumin, HbA1c, and serum creatinine—in predicting DR risk. These findings reinforce the clinical practice of monitoring kidney function and glycemic control as integral components of DR risk assessment, as these indicators have been shown to play a critical role in the progression of DR.

This study possesses several strengths that enhance its relevance and validity. The utilization of a large, balanced dataset comprising 3,000 diabetic patients provides a robust foundation for model training and validation, thereby reducing the risk of overfitting and improving the generalizability of the results. Furthermore, the study's comparative approach, which analyzes four distinct models, facilitates a comprehensive understanding of the strengths and limitations inherent to each model type in predicting diabetic retinopathy (DR). This approach not only underscores the high performance of the random forest and XGBoost models but also offers valuable insights into the clinical applicability of logistic regression and neural networks, particularly in contexts where specific interpretability or complexity requirements are paramount.

Moreover, this study employed multiple evaluation metrics—accuracy, precision, recall, F1-score, and AUC—providing a multidimensional assessment of each model's performance. This comprehensive evaluation facilitates more informed decision-making when selecting models for clinical implementation, ensuring that the chosen models satisfy both predictive and practical requirements. Additionally, the feature importance analysis enhances the study's utility by offering clinicians insights into the most predictive factors for diabetic retinopathy (DR), thereby guiding targeted monitoring of high-risk individuals.

However, this study also has limitations that warrant consideration. Firstly, the dataset was obtained from a single national health database, which may restrict the generalizability of the results to other populations with varying demographic and healthcare characteristics. Validation of these models on external datasets from diverse geographic and clinical settings is essential to confirm their robustness and applicability across different patient populations. Secondly, although this study compared multiple models, it did not examine ensemble techniques that integrate the strengths of various algorithms, such as stacking or blending [22–24]. Future research could explore ensemble methods, which may further enhance predictive performance by combining the advantages of multiple models within a single framework.

Another limitation is the risk of overfitting, particularly in complex models such as random forest and XGBoost, which are composed of numerous decision trees or parameters [25]. Although cross-validation was employed to mitigate this risk, overfitting continues to be a concern when these models are applied to new data [26]. Future studies could explore the use of regularization techniques or simplified model structures to enhance model robustness.

Additionally, the models relied on static clinical and biochemical data, failing to account for longitudinal trends in patient data over time. Incorporating time-series data in future research could enhance diabetic retinopathy (DR) risk predictions by capturing dynamic changes in key biomarkers, potentially leading to more accurate predictions of disease progression.

Additionally, the study did not incorporate other potential predictors of diabetic retinopathy (DR), such as genetic information, lifestyle factors, or advanced imaging data (e.g., retinal scans) [27]. The inclusion of these additional data sources could further enhance model accuracy and deepen our understanding of the factors contributing to DR. Future research should explore the integration of such multimodal data to create more comprehensive predictive models [28,29]. Moreover, the "black-box" nature of complex models, such as random forests, XGBoost, and neural networks, presents a challenge for clinical interpretability. Although these models demonstrated strong performance, their lack of transparency may limit their acceptance in clinical settings where interpretability is crucial. Future studies could address this issue by employing explainable AI methods, such as SHAP (Shapley Additive Explanations) or LIME (Local Interpretable Model-agnostic Explanations), to enhance model interpretability and foster clinician trust [30,31].

Building upon these findings, several avenues for future research are recommended. Validating these models across diverse datasets would help establish their generalizability, ensuring robust performance in various healthcare settings. Incorporating additional data types, such as genetic markers or retinal imaging, could further enhance prediction accuracy [32,33]. Furthermore, exploring ensemble models may allow for the integration of the strengths of logistic regression, random forest, XGBoost, and neural networks into a cohesive predictive framework. Future studies could also utilize longitudinal patient data to capture temporal patterns in diabetic retinopathy risk, potentially yielding more accurate insights into disease progression.

Exploring the cost-effectiveness and integration of these models into clinical workflows is essential for assessing their practical viability. While predictive accuracy remains critical, real-world implementation necessitates consideration of model deployment, ease of use, and alignment with existing healthcare infrastructure. Evaluating these factors in future research will ensure that these models deliver tangible benefits without overburdening healthcare providers.

This study provides robust evidence that both random forest and XGBoost are highly effective models for predicting diabetic retinopathy, thereby offering valuable tools for early detection and targeted intervention. The identification of renal and glycemic markers as key predictors underscores their clinical relevance and establishes a foundation for focused monitoring of high-risk individuals. While logistic regression and neural networks present complementary strengths—particularly in interpretability and the ability to capture non-linear patterns—random forest and XGBoost remain the most promising models for clinical application. Further research aimed at validation, interpretability, and integration into clinical practice will be essential for advancing the use of machine learning in diabetic retinopathy screening, ultimately improving patient outcomes and alleviating the burden of this sight-threatening condition.

## Supporting information

**S1 File. Raw data.**
(XLSX)

## Author contributions

**Conceptualization:** Bin Yang.

**Data curation:** Penglu Yang.

**Funding acquisition:** Bin Yang.

**Methodology:** Bin Yang.

**Writing – original draft:** Penglu Yang.

**Writing – review & editing:** Bin Yang.

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
