## [Decision Letter · Decision Letter 0]

17 Dec 2024

PONE-D-24-54045Development and Validation of Predictive Models for Diabetic RetinopathyPLOS ONE

Dear Dr. YANG,

Thank you for submitting your manuscript to PLOS ONE. After careful consideration, we feel that it has merit but does not fully meet PLOS ONE’s publication criteria as it currently stands. Therefore, we invite you to submit a revised version of the manuscript that addresses the points raised during the review process.

We look forward to receiving your revised manuscript.

Kind regards,

Sameena Naaz

Academic Editor

PLOS ONE

Journal Requirements:

2. Please include a caption for figure 2.

Additional Editor Comments :

Kindly respond to all the reviewer comments and submit an updated paper

Reviewers' comments:

Reviewer's Responses to Questions

**Comments to the Author**

1. Is the manuscript technically sound, and do the data support the conclusions?

Reviewer #1: Yes

Reviewer #2: Yes

2. Has the statistical analysis been performed appropriately and rigorously? 

Reviewer #1: Yes

Reviewer #2: Yes

3. Have the authors made all data underlying the findings in their manuscript fully available?

Reviewer #1: Yes

Reviewer #2: Yes

4. Is the manuscript presented in an intelligible fashion and written in standard English?

Reviewer #1: Yes

Reviewer #2: Yes

5. Review Comments to the Author

Reviewer #1: There are several issues present in the submitted manuscript. These issues have been listed as follows:

1. The authors have presented a comprehensive comparison approach, however, the authors fail to present more recent publications that support their comparative analysis. More recent publications should be incorporated too.

2. There are multiple Typo and grammatical mistakes which need to be corrected by the authors. Such as in line 283, there are two full stops.

3. The use of citations for the dataset used is not consistent with the citation style used in the rest of the document.

4. The quality of figures is bad and in some cases not readable. Also, for some reason, there is Fig2.pdf icon present at line 193. Kindly look into this issue.

5. Though the contents of the tables have been discussed, the figures provided have not. Kindly provided an extensive discussion regarding the result achieved for each figure.

Reviewer #2: 1. The paper is very well written in an organized manner

2. The paper fails to provide a comprehensive review and comparision with existing published work on Diabetic Retinopathy.

3. Recommended with minor addition

6. PLOS authors have the option to publish the peer review history of their article (what does this mean? ). If published, this will include your full peer review and any attached files.

**Do you want your identity to be public for this peer review?** For information about this choice, including consent withdrawal, please see our Privacy Policy .

Reviewer #1: **Yes: ** Muhammad Junaid Anjum

Reviewer #2: No

---

## [Author Response · Author response to Decision Letter 1]

30 Dec 2024

Dear Reviewers,

Thank you for providing the opportunity to revise our manuscript, titled “Development and validation of predictive models for diabetic retinopathy using machine learning”. We greatly appreciate the time and effort invested by the reviewers to improve the quality of our work. Below, we provide our point-by-point response to each of the reviewer’s comments.

Reviewer #1

1."The authors have presented a comprehensive comparison approach, however, the authors fail to present more recent publications that support their comparative analysis. More recent publications should be incorporated too."

To address this, we have incorporated more recent publications into the discussion and comparative analysis section. These include studies published in 2022–2024 that explore the application of machine learning models in diabetic retinopathy prediction and validate our findings. Relevant citations have been added to strengthen our analysis.

2."There are multiple Typo and grammatical mistakes which need to be corrected by the authors. Such as in line 283, there are two full stops."

We have thoroughly reviewed the manuscript to identify and correct all typographical and grammatical errors, including the redundant punctuation at line 283.

3."The use of citations for the dataset used is not consistent with the citation style used in the rest of the document."

We have corrected the formatting of the dataset citation to ensure consistency with the citation style used throughout the manuscript.

4."The quality of figures is bad and in some cases not readable. Also, for some reason, there is Fig2.pdf icon present at line 193. Kindly look into this issue."

We apologize for the oversight regarding the quality of the figures and the misplaced Fig2.pdf icon. The figures have been updated to high-resolution versions, and the icon has been removed. Additionally, figure legends have been revised for clarity.

5."Though the contents of the tables have been discussed, the figures provided have not. Kindly provide an extensive discussion regarding the result achieved for each figure."

Thank you for pointing this out. We have added an in-depth discussion of the results depicted in each figure, highlighting their relevance to the study’s findings and implications. The updated discussion is included in the results section.

Reviewer #2

1."The paper is very well written in an organized manner."

We thank the reviewer for their positive feedback and are glad that the manuscript's organization was appreciated.

2."The paper fails to provide a comprehensive review and comparison with existing published work on Diabetic Retinopathy."

We have addressed this by expanding the literature review and integrating a broader comparison with existing studies on diabetic retinopathy. The revised section now includes more detailed discussions on prior work and highlights how our findings align with or differ from these studies.

3."Recommended with minor addition."

We have incorporated the suggested additions and revisions based on the feedback provided, ensuring the paper is comprehensive and addresses all the reviewers' concerns.

We sincerely hope that the revised manuscript meets the reviewers' expectations. Thank you again for the opportunity to improve our work. Please let us know if further modifications are required.

Sincerely,

Bin Yang

---

## [Decision Letter · Decision Letter 1]

14 Jan 2025

Development and Validation of Predictive Models for Diabetic Retinopathy

PONE-D-24-54045R1

Dear Dr. YANG,

We’re pleased to inform you that your manuscript has been judged scientifically suitable for publication and will be formally accepted for publication once it meets all outstanding technical requirements.

Kind regards,

Sameena Naaz

Academic Editor

PLOS ONE

Additional Editor Comments (optional):

Reviewers' comments:

Reviewer's Responses to Questions

**Comments to the Author**

1. If the authors have adequately addressed your comments raised in a previous round of review and you feel that this manuscript is now acceptable for publication, you may indicate that here to bypass the “Comments to the Author” section, enter your conflict of interest statement in the “Confidential to Editor” section, and submit your "Accept" recommendation.

Reviewer #1: All comments have been addressed

Reviewer #2: All comments have been addressed

2. Is the manuscript technically sound, and do the data support the conclusions?

Reviewer #1: Yes

Reviewer #2: Yes

3. Has the statistical analysis been performed appropriately and rigorously? 

Reviewer #1: Yes

Reviewer #2: Yes

4. Have the authors made all data underlying the findings in their manuscript fully available?

Reviewer #1: Yes

Reviewer #2: Yes

5. Is the manuscript presented in an intelligible fashion and written in standard English?

Reviewer #1: Yes

Reviewer #2: Yes

6. Review Comments to the Author

Reviewer #1: The authors have addressed all of the comments given to them and have provided a more concrete research article.

Reviewer #2: Context of abstract should be highly clear and Compare the results of the paper with the existing paper

7. PLOS authors have the option to publish the peer review history of their article (what does this mean? ). If published, this will include your full peer review and any attached files.

**Do you want your identity to be public for this peer review?** For information about this choice, including consent withdrawal, please see our Privacy Policy .

Reviewer #1: **Yes: ** Muhammad Junaid Anjum

Reviewer #2: No

---

## [Editor Report · Acceptance letter]

PONE-D-24-54045R1

PLOS ONE

Dear Dr. Yang,

I'm pleased to inform you that your manuscript has been deemed suitable for publication in PLOS ONE. Congratulations! Your manuscript is now being handed over to our production team.

Kind regards,

on behalf of

Dr. Sameena Naaz

Academic Editor

PLOS ONE